# Methylation: An Ineluctable Biochemical and Physiological Process Essential to the Transmission of Life

**DOI:** 10.3390/ijms21239311

**Published:** 2020-12-07

**Authors:** Yves Menezo, Patrice Clement, Arthur Clement, Kay Elder

**Affiliations:** 1Laboratoire CLEMENT, Avenue d’Eylau, 75016 Paris, France; pclement@laboclement.com (P.C.); aclement@laboclement.com (A.C.); 2Bourn Hall Clinic, Bourn, Cambridge CB232TN, UK; kay.elder@gmail.com

**Keywords:** methylation, DNA, histone, epigenetics, gametes

## Abstract

Methylation is a universal biochemical process which covalently adds methyl groups to a variety of molecular targets. It plays a critical role in two major global regulatory mechanisms, epigenetic modifications and imprinting, via methyl tagging on histones and DNA. During reproduction, the two genomes that unite to create a new individual are complementary but not equivalent. Methylation determines the complementary regulatory characteristics of male and female genomes. DNA methylation is executed by methyltransferases that transfer a methyl group from S-adenosylmethionine, the universal methyl donor, to cytosine residues of CG (also designated CpG). Histones are methylated mainly on lysine and arginine residues. The methylation processes regulate the main steps in reproductive physiology: gametogenesis, and early and late embryo development. A focus will be made on the impact of assisted reproductive technology and on the impact of endocrine disruptors (EDCs) via generation of oxidative stress.

## 1. Introduction

Methylation is a universal biochemical process which covalently adds methyl groups to a variety of molecular targets, including neurotransmitters, lipids, proteins, and DNA. DNA repair, protein function, and gene expression involve methylation; it plays a critical role in two major global regulatory mechanisms: epigenesis and imprinting, which are transcriptional silencing and regulation of imprinted genes During reproduction, the two genomes that unite to create a new individual are complementary but not equivalent. For this reason, complete parthenogenesis (oocytes cleaving to form embryos without fertilization by sperm) or androgenesis (embryos developing without genetic contribution from an oocyte) cannot result in development of viable progeny. Methylation processes that regulate epigenesis and imprinting determine the characteristics of the regulatory processes that differ between male and female genomes. Imprinting “tags” a chemical mark in order to silence or activate genes, which is usually parent-specific. It can be reversible in order to switch genes “on” or “off” in different organs or during certain time periods (e.g., during pregnancy). This differs from epigenesis, which was defined by Conrad Waddington in 1942 as “causal interactions between genes and their products which bring the phenotype into being”. An epigenetic trait is a “stable heritable phenotype resulting from changes in a chromosome without alterations in the DNA sequence” [1] Modifications due to imprinting are considered more robust than those resulting from epigenesis. In general, both generate a unique chromosomal chemical modification for each parent, leading to different expressions of genes located on these chromosomes. Whether an imprinted gene is expressed depends upon the sex of the parent genome, with patterns of expression that result from chemical modification of DNA and/or alteration of chromatin structure via posttranslational modification of histone residues.

Epigenetic gene regulation is heritable and is directed by several different parameters, including DNA methylation, chromatin remodeling as a result of posttranslational histones modifications, and RNA interference. DNA methylation is executed by methyltransferases that transfer a methyl group from S-adenosylmethionine, the universal methyl donor, to cytosine residues of CG (also designated CpG) dinucleotides. In mammals, DNA methyltransferase-1 (DNMT1) is the predominant enzyme responsible for maintaining methylation during the cell cycle after DNA replication. De novo methylation, for example acquisition of new imprint tags, linked to the environment during pregnancy is principally carried out by DNMT3A and B: it will be necessary for epigenetic resetting in germ cells.

The importance of gene expression regulation via RNA interference (RNAi) must not be underestimated; however, this paper will focus on the methylation process and the accompanying biochemical pathways that affect these ubiquitous major effectors in the transmission of life.

### 1.1. A Reminder of the Methylation Process

S-adenosyl methionine (SAM) is the universal cofactor for methylation as the methyl group donor (Figure 1). Methionine adenosyltransferase (MAT) catalyzes SAM formation by linking methionine and ATP. Methylation is then carried out by DNA methyltransferase and histone methyltransferase enzymes. After the targets have been methylated, S-adenosyl homocysteine (SAH) is formed, and homocysteine (HCY) is then released. Hcy must be recycled: it is a toxic metabolite that inhibits the methylation process and can also inactivate some proteins via homocysteinilation, which leads to structural modifications. The one-carbon cycle is the major cellular pathway that recycles Hcy, together with methionine synthase associated with the folate cycle (see Figure 1). Methyl tetrahydrofolate (MTHF) is the active compound for folate cycle methionine synthase activity, and correct methylation depends upon a supply of MTHF. Some types of cells can also recycle Hcy via the cystine beta synthase (CBS) pathway, releasing cysteine. A third mechanism, available primarily in liver cells but only marginal in other cells, uses the betaine homocysteine pathway. Cysteine plays an important role in the redox homeostasis that is necessary for correct methylation, and errors in methylation are heavily linked to unbalanced oxidative stress. Abnormally low/inadequate concentrations of methionine and cystine may have an effect on epigenetic processes, and mechanisms for cellular transport and/or synthesis of methionine, cystine, glycine, and glutamic acid are crucial in supporting correct methylation. The availability of these compounds for direct (methionine) or indirect (via the synthesis of glutathione) participation in methylation as well as protection of the process are essential for correct establishment and maintenance of epigenetic and imprinting methyl tags.

### 1.2. The Beginning of Life: Fertilization and Immediately Postfertilization

During oogenesis, the oocyte accumulates epigenetic marks both on histones and on DNA, and this process continues until just before ovulation. At the time of fertilization, major intermediate metabolic changes are immediately initiated, which are necessary to support correct epigenetic/methylation status. All of the regulatory processes are strictly dependent upon maternal reserves of proteins and messenger RNAs (mRNAs) stored during oocyte growth; no transcription takes place in the newly fertilized egg until the zygote genome is activated (maternal to zygotic transition, MZT) at the 4- to 8-cell stages in human embryos. Interference RNAs (iRNAs) play an important role in regulating the translation of stored polyadenylated mRNAs.

Upregulation of the pentose phosphate pathway permits the formation of NADPH, essential for glutathione (GSH) synthesis: γ-L-Glutamyl-L-cysteinylglycine is a universal antioxidant molecule. GSH is necessary for sperm head swelling by opening the protamines that padlock the DNA and in order to protect the methylation process and to maintain redox homeostasis in order to prevent errors in methylation linked to unbalanced oxidative stress. As mentioned previously, abnormally low/inadequate concentrations of methionine and cystine may affect epigenetic processes [2]. Glutaredoxins are red-ox enzymes, classified as “light proteins”, that use glutathione as a cofactor. These proteins are thiol-disulfide oxidoreductases with a glutathione-binding site and one or two active cysteines in their active site. They can reduce methionine sulfone (oxidized methionine) to active methionine. Glutaredoxins are oxidized by oxidized substrates and are nonenzymatically reduced by reduced glutathione, which in turn is oxidized but can be regenerated by glutathione reductase (GRX). In terms of physio-biochemistry, glutaredoxins are present and highly expressed in early embryos: they actively protect redox homeostasis and thus have an impact on imprinting processes. Hypotaurine (Htau) is also found in the in vivo embryonic environment as an efficient antioxidant synthesized by oviductal cells and released into the tubal lumen [3,4]

Oocytes and early embryos have active mechanisms for cellular transport and/or synthesis of methionine, cystine, glycine, and glutamic acid. SAM is actively synthesized, and all of the enzymes involved in the methylation process are present, including the methionine synthase pathway that regenerates methionine from Hcy. The expression of CBS pathway enzymes is weak or absent, which means that homocysteine cannot be recycled to cystine, reinforcing the requirement for cystine and methionine. The betaine homocysteine pathway (BHMT) is only marginally represented. Human oocytes express high levels of folate receptor 1 and folate transporter1 (SLC19A1), indicating that these molecules play an important role during the first three to four days of development prior to the onset of genomic activation, MZT. Folates are central to a system that involves high molecular trafficking [5], and all of the enzymes involved in the folate and 1-carbon cycles are highly expressed in the oocyte. The embryo also finds important metabolites in its tubal fluid environment.

A major upheaval occurs during and immediately after the fertilization period: the DNA of the zygote genome is thought to be rapidly de-methylated immediately postfertilization, and human in vitro fertilized IVF embryos apparently follow this scheme. High sperm methylation and retained methylation of the paternal genome represent major markers of fertility [6,7,8]. The observation that isoforms affecting the one-carbon and folate cycles have a negative impact on fertility confirms the importance of a high methylation status in sperm. Paternal effects on early embryonic development is a feature that is well documented in bovine embryos. Suboptimal sperm methylation in terms of “quality and quantity” (CpG island regions restricted to retained histones) has a negative impact on human embryo blastocyst development [9] Approximately 10–15% of histones are retained, but these are not randomly located: they bind to the “active part/coding area” of DNA and so play an important role in embryonic development [6,9].

As mentioned earlier, global methylation in the sperm nucleus (DNA and histones) provides the key for correct spatial and biochemical conformation that will allow rapid access to the paternal genome after nuclear swelling in the sperm head [7,8]. This feature is mandatory for rapid S-phase activation and the major methylation/epigenetic modifications that will modify the male genome. Minor epigenetic alterations in sperm will immediately affect the early stages of preimplantation embryo developmental capacity. The paternal genome is rapidly demethylated initially, whereas the maternal genome is passively demethylated slowly during the subsequent cell cycle. The majority of the epigenetic methyl tags should be erased during de-methylation, but those that are retained and transmitted to the offspring probably transmit important epigenetic and imprint information. In human embryos, just prior to and during the second half of the pronuclear stage and before entering the first mitotic division, demethylated paternal DNA is immediately re-methylated, together with de novo H3-K9 tri-methylation [10]. A similar feature is observed in mouse, rabbit, and pig embryos. Oocytes have a significant endogenous pool of polyadenylated mRNAs coding for DNMT1 (DNA methyltransferase1), which is responsible for maintenance of methylation [3,5,11,12].

Global demethylation during preimplantation embryo development excludes imprinted genes: the parental imprints must be protected by DNMT1. The expression of DNMT3, responsible for de novo methylation, is weaker.

Two major issues must be highlighted: maternal age and the environment and the significant genetic impact of mutations in the one-carbon and folate cycles [13,14,15] The negative impact of abnormal methylation on bovine blastocyst formation has been documented [16]. Methylene tétrahydrofolate reductase (MTHFR) knockout decreases the number of blastocysts obtained, and those remaining have fewer cells in both the inner cell mass and the trophectoderm. Older females have a reduced abundance of DNA methyltransferases in MII oocytes, as the majority of important oocyte mRNAS are involved in regulation of homeostasis (redox, intermediate metabolism, etc.). Incorrect regulation of DNA methylation will further lead to inappropriate gene expression [17].

### 1.3. Problems Associated with Assisted Reproductive Technology

Several publications have indicated that babies born as a result of IVF procedures have different methylation patterns to those conceived naturally [13,18,19,20,21] and this is not related to intrinsic characteristics of the gametes or to the etiology of male and/or female infertility [2]. Controlled ovarian hyperstimulation (COH) is known to have a borderline negative effect, but there may be effects on implantation, placentation, and ultimately perinatal outcome [13,17]. Suboptimal in vitro culture conditions may have a negative impact on regulatory processes. A recent paper suggested that a rapid drop in methylation can be observed in human embryos shortly after fertilization [22]. This is based on data obtained from IVF embryos and may be an artifact of culture conditions [2,11,23]. Under natural conditions, the methylation status is stabilized by maintenance of methylation. Two recent papers have highlighted a significant stumbling block, in that there is no in vivo control for human embryos. A high incidence of four major imprinting disorders was reported in Japanese babies born after assisted reproduction technologies ART [19], and a second publication describes the risk of ART-imprinted disorders that are linked to epimutations [20]. Both reports suggest that these disorders may originate in the period immediately following fertilization under culture conditions in current use [20]. The possible source of these problems has been clearly advocated [2,11,23,24]; firstly, controlled ovarian stimulation increases the level of homocysteine (Hcy) in follicular fluid [25]. Hcy competes with methionine for the same amino acid transporter [26], which leads to abnormal accumulation of Homocysteine in the oocyte. Secondly, current culture media do not support correct maintenance of methylation, even in mouse embryos that have passed the “standard” Mouse Embryo Assay (MEA) for toxicity [24]. The majority of culture media lack both methyl donors and protection against methylation anomalies induced by oxidative stress. Methionine and cysteine are absent in some media [2,27]. Adding oviduct fluid to the medium might overcome these problems, but inclusion of methionine and cystine to support the one-carbon cycle could also provide a solution. Moreover, current culture media generate free radicals spontaneously [28]; these data highlight the fact that brief perturbation of the DNA methylation maintenance process in early stage embryos can influence development. In fact, DNA methylation, measured as the quantity of methylated CpG per embryo, does not decrease significantly until the time of maternal to zygotic transition and immediately after [29] (Figure 2). Errors in methylation may further impact placental function [30]. Phospholipid formation, another biochemical parameter linked to methylation, is also impaired by IVF and could affect the health of offspring [31].

### 1.4. Blastocyst Formation, Implantation, and Placental Development

Initial differentiation of cells in the inner cell mass and trophectoderm is accompanied by differential methylation. Maternal DNA methylation regulates early trophoblast development [33], and miscarriages may be attributed to errors in differential methylation. Methylation that accompanies zygotic genomic imprinting affects further placental development [34]. In the male, DNA methylation has an effect on trophoblastic function in general, and methylation errors will therefore have a negative impact [35]. Defective methylation affects overall trophoblast development and its physiological capacity to sustain implantation and, subsequently, placental development and fetal growth [33,36].

Trophoblastic tissue remains hypomethylated relative to embryonic tissues. Alterations in the cellular adhesion profile of trophoblast cells are regulated by methylation of maternal DNA [33]. Imprinting appears to be particularly important both for placental and for embryo development; the development of extraembryonic tissues is under paternal control, whereas embryo growth is directed by maternal genes. For this reason, although cloning by duplication of pronuclear paternal or maternal genomes is possible, it can never progress to a viable outcome, even if implantation takes place. Mouse androgenotes display hypertrophy of the placenta (placentomegaly), and the opposite is observed in gynogenesis. The “sex conflict hypothesis” could be linked to imprinting mechanisms: The *maternal goal* is to not invest all of her resources in a single offspring and to limit fetal use of maternal resources. The *paternal goal* is to maximize the size of an individual by maximizing fetal use of maternal resources. However, numerous exceptions to this hypothesis are evident, as demonstrated by IGF2/H19 expression. Paternally expressed *IGF2,* which is highly expressed in normal mouse and human placenta, regulates placental growth via growth factor activity and nutrient permeability. The gene for maternally expressed *H19* lies on the same chromosome as the IGF2 gene, and it allows the synthesis of a 2.3 Kb noncoding RNA (NcRNA). Hypermethylation of the *H19* promoter on the paternal allele allows for expression of the paternal allele for IGF2 and decreases (represses) H19 expression. IGF2 drives placental growth as well as the capacity to transfer nutrients to the fetus Maternally expressed *H19* genes suppress growth, and expression of *H19* is continuously regulated (biallelic to monoallelic and reverse) throughout embryonic development. However, anomalies linked to aberrant placental IGF2 methylation lead to fetal growth restriction [37]. The balance between IGF2 and H19 expression is of major importance for viable fetal gestation, and monoallelic expression can occur in the absence of imprinting.

The placenta is the most hypomethylated DNA in human tissue, and DNA methylation increases specifically with gestational age [38], coupled to a high demand for methionine in the third trimester of pregnancy [39]. A striking observation confirms the link between elevated circulating homocysteine, oxidative stress, and methylation [40,41,42]. Homocysteine is both a cause and a consequence of oxidative stress, which can lead to pre-eclampsia [43] A further observation has shown that the oxygen load delivered during parturition in order to stabilize preterm babies affects their methylome. It is not known whether the process is reversible, and the authors [44] suggest that this could affect the response to oxidative stress, DNA repair, and cell progression. Ultimately, placental growth is intricately linked to DNA methylation [40], and correct homeostasis to allow effective methylation avoids adverse pregnancy outcomes such as pre-eclampsia [41,42,43,44,45,46].

### 1.5. Embryo Growth, DNA Methylation, and Fetal Testis and Ovary

DNA methylation plays a major role throughout embryo growth; however, tracking the sequence of variations is very complex due to the fact that epigenetic alterations will modify both gene and promoter methylation, leading to epipolymorphism [41]. All of the methylation/epigenetic processes can be modified by the environment, and modifications can be transferred to the next generation via gametogenesis. This feature has now been clearly established [47,48,49]. The progeny of female rats injected with plastic endocrine disruptors during pregnancy shows severe pathologies for at least 3 generations, with altered germline epigenome transmission observed between generations. Experiments carried out with the yellow agouti mouse model demonstrated that the results involved methylation. The deleterious impact of endocrine disruptors and other xenogenic chemical compounds has been clearly demonstrated [42,47,48,49,50,51]. Supporting the one-carbon cycle with appropriate exogenous compounds can rescue normal gametogenesis, and thus, the impact appears to be clearly due to erratic methylation [51,52,53]. Fetal ovaries and testes are very sensitive to environmental perturbation, including food, air pollution, and nanoparticles. DNA methylation and gametogenesis are intricately linked due to the fact that primordial germ cells are profoundly demethylated and subsequently re-methylated during a later developmental period: prenatal life in males and postnatal development in females. For this reason, the environmental impact on the testis is more significant [54,55], as resetting of methylation may be impaired via toxicants carried by maternal blood and amniotic fluid. The fetal testis is a major target for endocrine disruptors such as herbicides, pesticides, and PolychloroBisphenyls PCBs, and determining whether the anomalies have occurred during fetal life or during the prepuberal period can be complicated. The effects can include inducing low sperm count, testicular cancer, cryptorchidism, undescended testis, ambiguous genitalia, and “testis dysgenesis syndrome (TDS)”: environmental exposure is the primary factor involved in the phenotypes associated with this syndrome. The adult endocrine system may be affected either directly or indirectly via epigenetic mechanisms. The genetic possibility of errors cannot be totally excluded, but the consensus is in favor of an environmental problem affecting epigenesis and methylation [2,47,48,49,54,55,56].

### 1.6. Methylation Errors in Gametogenesis, Postnatal Life, and Adults

Numerous aspects of modern lifestyle have contributed to an increase in life expectancy since World War II. However, we are now faced with a downturn that seems to be related to environmental issues. This paradigm parallels an increased prevalence of problems associated with fertility: decreased sperm quality, an increase in premature ovarian failure, and diminished ovarian reserve syndromes. The decline in male fertility is no longer a matter for debate and has been linked to the effect of environmental oxidative stress on methylation. As mentioned earlier, resetting of DNA methylation in germinal cells is initiated during fetal life and the fetal testis is more sensitive than the fetal ovary to exogenous (chemical) toxins carried in maternal blood. In women, the scarcity of material for study makes the situation difficult to analyze; damage arises over a short period of time during oocyte maturation, but the problems may start earlier, during “quiescent periods”, when the oocytes have no protection against exogenous hazards [57].

### 1.7. Methylation Errors in the Male

During spermatogenesis, DNA methylation is active during spermatogonial mitosis and meiosis and less active during spermiogenesis [58]. Histone methylation has an immediate effect during fertilization; it exerts a positive effect on the embryonic S-phase and on subsequent preimplantation development [7,8]. It is now recognized that pathologies can be transmitted across generations via epimutations that occur in adult gametogenesis [59,60,61,62,63]. Oxidative stress plays a pivotal role in this process, with glucose, lipids, and homocysteine as well as chemicals at the center of the problem. Diabetic patients have a high level of oxidative stress due to excess ROS generated by glucose metabolism at the level of the Krebs cycle (succinyl dehydrogenase) and inhibition of hypoxanthine-guanine phosophoribosyltransferase (HPRT). Although epigenetic disorders can be transmitted via sperm [6,60,63], ROS may also affect the primary sperm DNA structure directly via the formation of DNA adducts such as 8-oxo deoxyguanosine: all of the nuclear bases may be affected by oxidation, but guanine is the most sensitive. The oocyte has a redundant but finite capacity for DNA repair, which decreases with maternal age [57,64]. Interestingly, it has been shown that oxidized sperm also has an effect on epigenetic reprogramming of the preimplantation embryo [65]. Zygotic DNA repair machinery (base excision repair) is activated at the expense of active demethylation of paternal DNA, another unexpected observation that links oxidative stress to errors in methylation [55,56]. In obese patients, peroxidized lipids may oxidize sperm DNA and membranes. In the case of ART and intracytoplasmic sperm injection (ICSI) in particular, injected oxidized membrane lipids and DNA can induce oxidative stress (OS) in the oocyte. Since spermatogenesis is a continuous process, it is difficult to determine at what point the problems might begin [66]; fortunately, therapeutic treatments can easily cover a complete cycle of Tspermatogenesis [15,53,67] The genetic background of MTHFR SNPs (A1298C and C677T) significantly perturbs the sperm methylome [15,68]: patients who carry these polymorphisms are not able to metabolize synthetic folic acid, with accumulation of unmetabolized folic acid (UMFA). This must be taken into consideration to avoid further damage to the sperm methylome [68].

Plastic endocrine disruptors (EDCs) exert a transgenerational effect: they induce oxidative stress (Figure 3) via the estrogen receptor (ER), proliferation of activated peroxisome receptors and constitutive androstane receptor (CAR), pregnan X receptor (PXR), and aryl carbon (Ah) receptors.

The underlying process has been elucidated and demonstrates an immediate impact on methylation [55]. When cytosine is oxidized at the level of differentially methylated regions (DMRs) in DNA, CpG islands, and/or imprinting control regions (ICRs), there is a risk of interference with regulatory processes by demethylation. Oxidative stress affects histone acetylation and de-acetylation [69], with indirect perturbation of “the epigenetic landscape”—methylation tags are transmitted to the offspring. In general, these posttranslational modifications (PTMs) act as good markers of sperm quality. Methylation occurs during spermatid elongation, and this is the most important PTM. The biochemical process is driven by methyltransferases that are specific to Arginine (R) or Lysine. (K), with S-AdenosylMethionine (SAM) as a co-effector. Decreased methylation on H3K will lead to disorganized spatiotemporal embryo development. These positioned methyl tags can regulate transcription by either repressing or increasing gene expression. The spermatozoon allows ancestral history to be transmitted via methylation, as demonstrated by the clear example of the 1944–1945 Dutch Famine. Individuals exposed prenatally to food shortage were found to have reduced methylation status of the imprinted IGF2 gene several decades later. Together with methylated DNA, methylated histones carry epigenetic and imprinting marks which will modulate phenotypic and biochemical variations. Sperm histones/DNA methylation should be considered as a means of transmitting acquired heredity, and this finely tuned system can be impaired by environmental factors.

Methylation anomalies, either hypomethylation or hypermethylation depending upon the target (genes or their promoters), may lead to testicular cancer, initiated from the period of gonadal development. However, as with other genital tract cancers, the true impact of methylation anomalies is not proven.

### 1.8. Methylation in Female Gametes

Methylation undergoes a major resetting during follicular growth and oocyte maturation. CpG methylation is largely established in the immature germinal vesicle stage of the oocyte. Mitochondrial DNA (mtDNA) methylation does not occur in the maturing oocyte [65]. As the oocytes mature, there is a net accumulation of methylation at both CpG and non-CpG targets [70] as well as a net gain in histone methylation (H3K9 and H3K4) [71]. Superovulation alters DNA methylation, but the impact seems to be borderline. However controlled ovarian hyperstimulation increases follicular fluid Hcy, with toxic effects on oocyte quality [25,72,73]. In vitro maturation alters the oocyte epigenom, due to the fact that culture conditions are far from optimal. Any interference with methylation processes represents a possible source of long-life pathologies, highlighting the potential risks of ART.

The impact of oxidative stress originating from EDCs [47,48,49,74], maternal obesity, and diabetes that is seen in males is also true for women. OS alters oocyte DNA, which must be repaired during the fertilization period. Experience from ART has demonstrated that OS decays are shared equally between males and females [75]; DNA repair is achieved at the expense of methylation. However, maternal DNA is passively demethylated during preimplantation embryo development. Once again, it is difficult to determine when the problem arises: fetal life, prepubertal life, or during sexual maturity [47]. EDCs cross the placenta, and it should be remembered that they affect mitochondrial function, which is very sensitive to OS: this in itself generates free radicals, and mitochondria are maternally transmitted. The majority of chronic female pathologies are linked to the relationship between oxidative stress and methylation: this is true for PCOS [76,77], which is characterized by elevated oxidative stress, insulin resistance, and more significantly, elevation of circulating homocysteine [25]. The same observation has been made for endometriosis, although the mechanisms are not clear and remain controversial [78]. The contribution of histone methylation to endometrial carcinogenesis seems to be important [79] and may also be featured in the pathology of premature ovarian failure (POF), ovarian resistance, and early menopause. EDCs have structural similarities to estrogen, and the majority can interact first directly with the ovary and secondly with cellular estrogen receptors to potentially induce OS. In addition, nanoparticles travel between different organs, passing through the ovarian circulation barrier to induce OS; the extent of toxicity depends upon their composition [80].

DNA and histones methylation and the associated epigenetic modifications also promote endometrial and ovarian cancer [79,81]. Epithelial ovarian cancer (EOC) is the most frequent of the heterogeneous group of ovarian cancers. MicroRNAs (miRNAs) appear to be major inducers of epigenetic anomalies [82], but the importance of DNA methylation, together with anomalies of chromatin remodeling, must be acknowledged.

## 2. Conclusions

Methylation is a first-line essential biochemical process in the transmission of life, playing a critical role in modification of DNA and histones. It is involved in regulating gametogenesis, embryonic and placental growth, as well as imprinting and epigenesis. Inhibition by miRNA should be considered a second-line level of regulation. Gametes transfer not only DNA but also information for the regulation of early and late embryo development as well as ancestral history, skills, and endocrinology. However, this precise, finely tuned system is now increasingly subject to impairment by environmental factors, including endocrine disruptors and generators of oxidative stress in general [47,48,49,50,51,52,53,54,55,56]. Dysregulation of methylation processes (histones and DNA) leads to cancer. Dietary supplements that avoid synthetic folic acid and support the one-carbon and the folate cycles in both men and women have positive benefits and improve fertility [15,51,52,53,67], especially for carriers of the MTHFR SNPs. This type of dietary supplementation should be proposed to all patients initiating an ART protocol in order to increase the store of “efficient folates” that will support effective metabolism in oocytes and to improve the sperm DNA methylome. Methylation is necessary for the synthesis of catecholamines, and SAM is a precursor of the biogenic amines (spermine, spermidine, and putrescine), multifunctional compounds involved in cellular growth. The one-carbon cycle is a permanent effector in regulation and dysregulation of all of the processes leading to transmission of life [83]. Methylation maintenance is a critical requirement during the very early stages of embryonic development and should be recognized in the design and definition of embryo culture media in assisted reproduction technologies [2].

## Figures and Tables

**Figure 1 ijms-21-09311-f001:**
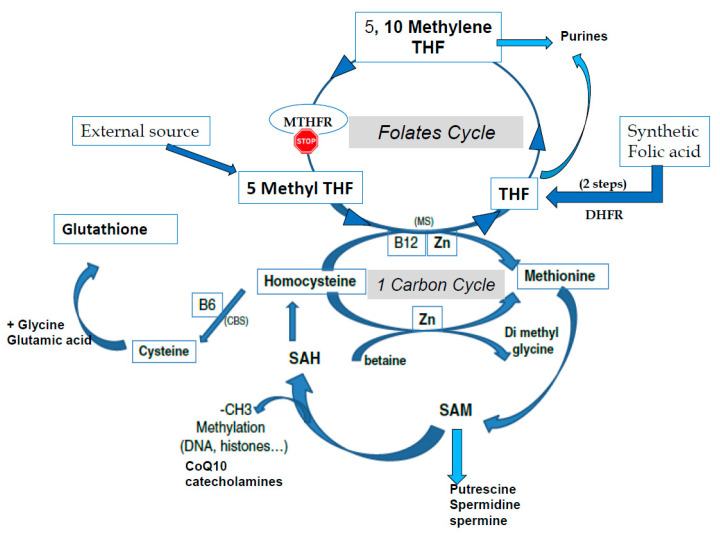
The one-carbon cycle (1-CC) and the folates cycle (MTHFR: methyltetrahydrofolate reductase, THF tetrahydrofolate). CBS: Cystathionine beta synthase, CoQ10: Coenzyme Q10, CH3: methyl group, DHFR: Dihydrofolate reductase, MS: methionine synthase, SAM S Adenosyl Methionine, SAH: S Adenosyl Homocysteine, Zn: Zinc.

**Figure 2 ijms-21-09311-f002:**
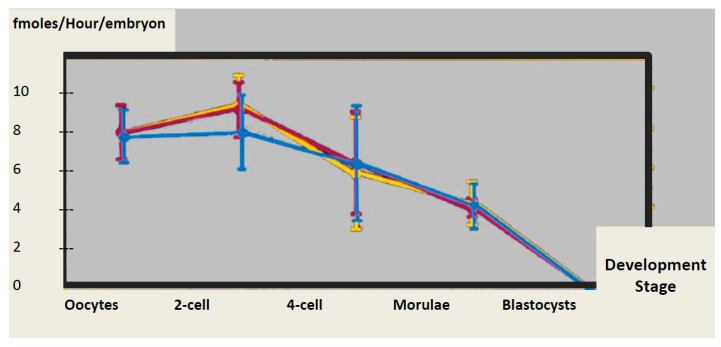
DNA methyl transferase activity in the mouse preimplantation embryo [32]. Yellow: caryogames, Blue: Ethanol activated parthenogenotes, Pink: Calcium ionophore activated parthenogenots.

**Figure 3 ijms-21-09311-f003:**
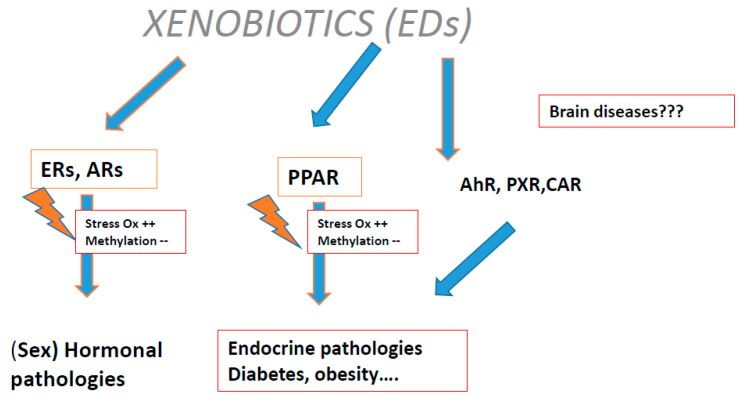
The link between oxidative stress and methylation disturbance: Xenobiotics (endocrine disruptors (EDCs)) interactions with some receptors.

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
