# Peer review of "Methylation: An Ineluctable Biochemical and Physiological Process Essential to the Transmission of Life"

_ijms, 2020, doi:10.3390/ijms21239311_

Round 1

Reviewer 1 Report

The present review manuscript focus on the methylation processes during gametogenesis and early and late embryo development as well as on the impact of assisted reproductive technology and on the endocrine disruptors (EDCs) via the generation of oxidative stress. As it has been mentioned methylation is a principal biochemical process in the transmission of life, playing a critical role in modification of DNA and histones. Moreover, the effect of ARTs on methylation process at early stages of embryo development is very important with consequences on adult life.

It is clear, well written and very comprehensive review covering all major parts of the methylation process and the related biochemical pathways based on the authors extensive experience on the subject and current knowledge in the literature. I enjoyed reading this paper and I believe more will do.

Minor comments:

Line 12: Delete 2nd full stop

Line 19: Correct “early” and “will”

Line 41: Delete full stop

Line 107: “and”

Line 268: Full stop after references (51,52).

Lines 358-360: Define sentence(s)

Author Response

1-The present review manuscript focus on the methylation processes during gametogenesis and early and late embryo development as well as on the impact of assisted reproductive technology and on the endocrine disruptors (EDCs) via the generation of oxidative stress. As it has been mentioned methylation is a principal biochemical process in the transmission of life, playing a critical role in modification of DNA and histones. Moreover, the effect of ARTs on methylation process at early stages of embryo development is very important with consequences on adult life.
It is clear, well written and very comprehensive review covering all major parts of the methylation process and the related biochemical pathways based on the authors extensive experience on the subject and current knowledge in the literature. I enjoyed reading this paper and I believe more will do.
Minor comments:
Line 12: Delete 2nd full stop DONE
Line 19: Correct “early” and “will” DONE
Line 41: Delete full stop DONE
Line 107: “and” DONE
Line 268: Full stop after references (51,52). DONE
Lines 358-360: Define sentence(s) DONE

Reviewer 2 Report

The review by Menezo et al analyzes the epigenetic reprogramming and, in particular DNA methylation in a perspective that is relevant to Assisted Reproductive Technology. This approach is interesting, however some statements are not substantiated by clear solid scientific evidences. Often the necessary citation of scientific works are missing.

There are typos all over the review. Numerous references are missing.

After a critical revision of the text and the update of the cited literature the review might be accepted.

The quality of Figure should be improved.

Below the main points that should be revised

Lines 12- 13: it plays a critical role in the two major global regulatory mechanisms, epigenesis and imprinting.

In this context, the Authors are referring to DNA or histone methylation

The term epigenesis is not in the scientific use. It relates to a phylosophic debate. They might use “epigenetic modification”, instead.

The sentence should be changed accordingly.

Line 14-15: Methylation determines the characteristics that differ between male and female genomes.

The phenomenon regards genomes from male and female gametes. Sentence should be modified accordingly

Lines 18-19: The methylation processes will be developed for the main steps of reproductive physiology

Without considering the typos, the sentence in not clear

Lines 27-28: it plays a critical role in the two major global regulatory mechanisms, epigenesis and imprinting.

the two mechanisms are:

transcriptional silencing of genes and regulation of expression of imprinted genes.

The sentence should be changed accordingly.

Line 32-33: Methylation determines the characteristics that differ between male and female genomes.

The phenomenon regards genomes from male and female gametes. Sentence should be modified accordingly

Lines 33-34: Imprinting ‘tags’ a chemical mark in order to silence or activate genes, and is usually parent-specific.

The sentence is unintelligible. If the Authors are referring to imprinting marks the sentence should be changed accordingly

Line 36, line 40: epigenesis

"Whether or not early mammalian development results from preformation or epigenesis remains an unresolved issue. Evidence for or against either is weak, inconclusive, and often misinterpreted. Yet, one can parsimoniously conceptualize formation of the mouse blastocyst as a series of random, stochastic events stemming from initial and sequential small asymmetries in egg, zygote, and cleavage stages. Differential compartmentalized gene expression does not lead but follows the morphogenesis and cell fate allocation in the mammalian blastocyst." from Solter D. Preformation Versus Epigenesis in Early Mammalian Development. Curr Top Dev Biol. 2016;117:377-91.

Line 42: Whether or not a gene is expressed depends upon the sex of the parent genome,

This is true for imprinted genes during embryo development. The sentence should be changed accordingly

Line 45: Epigenetic gene regulation is heritable

Epigenetic gene regulation is inheritable during mitosis from one cell to its daughter cells. It is not inheritable from one generation to a second or third generation. This is because during gametogenesis all the epigenetic setting is changed. In germ cell DNA methylation is high. Post-zygotically DNA methylation decreases till the blastocyst stage. After that it increases in somatic cells till cells are fully differentiated. After that DNA methylation is stable.

This concept should be specified in the text.

Lines 50-52: De novo methylation, for example acquisition of new imprint tags linked to the environment during pregnancy, is principally carried out by DNMT3A and B.

This sentence should be changed. De novo DNA methylation is fundamental for establishing the proper epigenetic setting in germ cells, in embryo, through development.

Line 53 : “The importance of regulation”

It is this sentence referring to regulation of gene expression?

Line 53, line 86: interference RNA (iRNA)

The correct term is RNA interference (RNAi)

Lines 99-101: glutaredoxins are present and highly expressed in early embryos: they actively protect redox homeostasis, and thus have an impact on imprinting 100 processes.

I found no literature on the role of glutaredoxin in imprinting.

Lines 122-123: Approximately 15% of histones are retained, but these are not randomly located: they are methylation sites,

The sentence is not clear. The meaning of “methylation sites” should be explained. References should be cited.

Lines 124-126: Global methylation in the sperm nucleus provides the key for the correct spatial and biochemical conformation that will allow rapid access to the paternal genome after nuclear swelling in the sperm head.

This concept has been introduced at line 90. It might be useful to remind and connect the two information.

Line 130: the methyl tag

The meaning of “methyl tag” clarified. Usually the term "epigenetic tags" is used.

Line 132: Do the Authors mean “human zygotes” instead of “human embryos”?

Line 134: Reference 6 is old. Is there any updated reference to refer to?

Line 137: References 3,7 are self-citations. Is there any reference from other authors that support the statement ?

Line 139: the caption of Figure 2 is missing.

Line 177: as measured by the quantity of CpG per embryo,

Do the Authors mean “measured as quantity of methylated CpG per embryo”?

Line 182: the caption of Figure 3 is missing.

Line 226: “growth” should be substituted with “development”.

Line 230: This feature has now been clearly established

This is an overstatement.

Line 235 References 44,45

These are self-citations. Is there any reference from other authors? Additional citations would be useful.

Lines 238-241: DNA methylation and gametogenesis are intricately linked, due to the fact that primordial germ cells are profoundly demethylated and subsequently re-methylated during a later developmental period: prenatal life in males, and postnatal development in females.

The mature sperm genome shows 80%–90% overall CpG methylation, the highest global DNA methylation level of any cell in the mouse (Popp et al. 2010, Nature 463: 1101–1105.), yet the paternal genome is apparently completely demethylated shortly after zygote formation (Mayer et al 2000, Nature 403: 501–502).

Lines 243-245 The fetal testis is a major target for endocrine disruptors such as herbicides, pesticides and PCBs, and determining whether the anomalies have occurred during fetal life or during the prepuberal period can be complicated.

Which anomalies the Authors are referring to? References are missing

Lines 245-248 The effects can include inducing low sperm count, testicular cancer, cryptorchidism, undescended testis, ambiguous genitalia, "testis dysgenesis syndrome (TDS)": environmental exposure is the primary factor involved in the phenotypes associated with this syndrome.

All these features can be due to genetic defects.

Environmental exposure to mutagens can induce germinal DNA mutations regarding either small DNA sequences or gross chromosomal changes.

There is a higher rate of mutation in spermatogenesis in comparison to oogenesis. This is due to the high number of mitosis occurring in male gametogenesis.

The sentence should be rephrased.

Line 252: “faced with” should be substituted with “facing”

Line 253: “is paralleled with” should be substituted with “parallels”

Line 257: “methylation tags “ should be substituted with “DNA methylation”

Lines 261-262: For example, major methylation anomalies are observed in Turner and Klinefelter syndromes, although this may be an independent

This sentence should be modified or removed. The problem in these syndromes is the chromosome X aneuploidy. The aneuploidy causes no pairing or anomalous sex chromosome pairing at meiosis, which interferes with the correct progression of gametogenesis.

Lines 268-269: It is now recognized that pathologies can be transmitted across generations via epimutations that occur in adult gametogenesis.

This is an overstatement.

Lines 276-277: The oocyte has a redundant but finite capacity for DNA repair, which decreases with maternal age.

This is not proven. There are significantly fewer DNA mutation in oogenesis than in spermatogenesis.

Lines 290-292: Plastic EDCs exert a transgenerational effect: they induce oxidative stress (figure 4) via the estrogen receptor (ER), proliferation of activated peroxisome receptors and constitutive androstane receptor (CAR), pregnan X receptor (PXR) and aryl carbon (Ah) receptors.

This statement should be supported by experiments or epidemiology, or both.

Lines 307-308: The spermatozoon allows ancestral history to be transmitted via methylation, as demonstrated by the clear example of the 1944-45 Dutch Famine.

On this topic there are other opinions. There is no demonstration of changes in DNA methylation in spermatozoa. The sentence should be modified.

Lines 308-309: Individuals exposed prenatally to food shortage were found to have reduced methylation status of the imprinted IGF2 gene several decades later.

This has been related to the intrauterine environment. The sentence should be modified.

Lines 314-316: Methylation anomalies, either hypomethylation or hypermethylation, depending upon the target (genes or their promoters), may lead to testicular cancer, initiated from the period of gonadal 315 development.

This is an overstatement. Methylation changes in cancer are known and are related to cancer development in general. The sentence should be modified.

Lines 111-112: Folates are central to a system that involves high molecular trafficking [3]

Other references are necessary

References should be introduced for sentences at lines 88-92, 103-104, 106-108, 108, 112-113, 117-119, 119-120, 126-127, 127-128, 128-130, 161-162, 230-232.

Author Response

The review by Menezo et al analyzes the epigenetic reprogramming and, in particular DNA methylation in a perspective that is relevant to Assisted Reproductive Technology. This approach is interesting, however some statements are not substantiated by clear solid scientific evidences. Often the necessary citation of scientific works are missing.
There are typos all over the review. Numerous references are missing.
After a critical revision of the text and the update of the cited literature the review might be accepted.
The quality of Figure should be improved.
Below the main points that should be revised
Lines 12- 13: it plays a critical role in the two major global regulatory mechanisms, epigenesis and imprinting. DONE
In this context, the Authors are referring to DNA or histone methylation YES
The term epigenesis is not in the scientific use. It relates to a phylosophic debate. They might use “epigenetic modification”, instead.
The sentence should be changed accordingly. DONE
Line 14-15: Methylation determines the characteristics that differ between male and female genomes.
The phenomenon regards genomes from male and female gametes. Sentence should be modified accordingly
Lines 18-19: The methylation processes will be develpped for the main steps of reproductive physiologie
Without considering the typos, the sentence in not clear MODIFIED
Lines 27-28: it plays a critical role in the two major global regulatory mechanisms, epigenesis and imprinting.
the two mechanisms are:
transcriptional silencing of genes and regulation of expression of imprinted genes.
The sentence should be changed accordingly. DONE
Line 32-33: Methylation determines the characteristics that differ between male and female genomes.
The phenomenon regards genomes from male and female gametes. Sentence should be modified accordingly
Lines 33-34: Imprinting ‘tags’ a chemical mark in order to silence or activate genes, and is usually parent-specific.
The sentence is unintelligible. If the Authors are referring to imprinting marks the sentence should be changed accordingly
Line 36, line 40: epigenesis
"Whether or not early mammalian development results from preformation or epigenesis remains an unresolved issue. Evidence for or against either is weak, inconclusive, and often misinterpreted. Yet, one can parsimoniously conceptualize formation of the mouse blastocyst as a series of random, stochastic events stemming from initial and sequential small asymmetries in egg, zygote, and cleavage stages. Differential compartmentalized gene expression does not lead but follows the morphogenesis and cell fate allocation in the mammalian blastocyst." from Solter D. Preformation Versus Epigenesis in Early Mammalian Development. Curr Top Dev Biol. 2016;117:377-91.
Line 42: Whether or not a gene is expressed depends upon the sex of the parent genome,
This is true for imprinted genes during embryo development. The sentence should be changed accordinglyDONE
Line 45: Epigenetic gene regulation is heritable
Epigenetic gene regulation is inheritable during mitosis from one cell to its daughter cells. It is not inheritable from one generation to a second or third generation. This is because during gametogenesis all the epigenetic setting is changed. In germ cell DNA methylation is high. Post-zygotically DNA methylation decreases till the blastocyst stage. After that it increases in somatic cells till cells are fully differentiated. After that DNA methylation is stable.There is a total removal then a reassessment of the methyl tags
This concept should be specified in the text. DONE
Lines 50-52: De novo methylation, for example acquisition of new imprint tags linked to the environment during pregnancy, is principally carried out by DNMT3A and B.
This sentence should be changed. De novo DNA methylation is fundamental for establishing the proper epigenetic setting in germ cells, in embryo, through development .DONE but timing of re setting will depend on the sex and notnecessarily done during pregnancy
Line 53 : “The importance of regulation”
It is this sentence referring to regulation of gene expression? DONE
Line 53, line 86: interference RNA (iRNA)
The correct term is RNA interference (RNAi) DONE
Lines 99-101: glutaredoxins are present and highly expressed in early embryos: they actively protect redox homeostasis, and thus have an impact on imprinting 100 processes.
I found no literature on the role of glutaredoxin in imprinting.See the paper concerning the protection of methylation in early embrry. Glutaredoxins is able of reduced the met sulfone to met… and so their role and the interplay with glutathione is of major importance. See also glutathione and sperm head sewelling
The following paper is one of the most downloaded Oxidative stress and alterations in DNA methylation: two sides of the same coin in reproduction.Menezo YJ, Silvestris E, Dale B, Elder K.Reprod Biomed Online. 2016 Dec;33(6):668-683
And this one axpalining the role of redoxines and the impact on methylation protection Epigenetic remodeling of chromatin in human ART: addressing deficiencies in culture media. Ménézo Y, Elder K.J Assist Reprod Genet. 2020 Aug;37(8):1781-1788

Lines 122-123: Approximately 15% of histones are retained, but these are not randomly located: they are methylation sites,
The sentence is not clear. The meaning of “methylation sites” should be explained. References should be cited. Done
Ihara M, Meyer-Ficca ML, Leu NA, Rao S, Li F, Gregory BD, Zalenskaya IA, Schultz RM, Meyer RG Paternal poly (ADP-ribose) metabolism modulates retention of inheritable sperm histones and early embryonic gene expression.
.PLoS Genet. 2014 May 8;10(5):e1004317.
Most relevant vs the early embryonic development

Lines 124-126: Global methylation in the sperm nucleus provides the key for the correct spatial and biochemical conformation that will allow rapid access to the paternal genome after nuclear swelling in the sperm head.
This concept has been introduced at line 90. It might be useful to remind and connect the two information.OK DONE
Line 130: the methyl tag
The meaning of “methyl tag” clarified. Usually the term "epigenetic tags" is used. DONE
Line 132: Do the Authors mean “human zygotes” instead of “human embryos”? Preimplantation embryo before MZT
Line 134: Reference 6 is old. Is there any updated reference to refer to? It is the best animal modelwith ewe and goat to compare to human Mono ovulatory , time to reach MZT
Line 137: References 3,7 are self-citations. Is there any reference from other authors that support the statement ?. Theree is another one from Benkhalifa et al. in Fert Steril but from our team
Dynamic expression of DNA methyltransferases (DNMTs) in oocytes and early embryos.Uysal F, Akkoyunlu G, Ozturk S.Biochimie. 2015 Sep;116:103-13

We will add this one
Line 139: the caption of Figure 2 is missing. Done
Line 177: as measured by the quantity of CpG per embryo,
Do the Authors mean “measured as quantity of methylated CpG per embryo”?
Line 182: the caption of Figure 3 is missing. DONE Line 226: “growth” should be substituted with “development”. DONE
Line 230: This feature has now clearly established
This is an overstatement.authors explainations
Line 235 References 44,45 but also 41 abd the erfe Mannikam et al. Cooney…. 1 CC support allows some correction: methylation process. Transgenrational passage
These are self-citations. Is there any reference from other authors? Additional citations would be useful.
Lines 238-241: DNA methylation and gametogenesis are intricately linked, due to the fact that primordial germ cells are profoundly demethylated and subsequently re-methylated during a later developmental period: prenatal life in males, and postnatal development in females.
The mature sperm genome shows 80%–90% overall CpG methylation, the highest global DNA methylation level of any cell in the mouse (Popp et al. 2010, Nature 463: 1101–1105.), yet the paternal genome is apparently completely demethylated shortly after zygote formation (Mayer et al 2000, Nature 403: 501–502).
Lines 243-245 The fetal testis is a major target for endocrine disruptors such as herbicides, pesticides and PCBs, and determining whether the anomalies have occurred during fetal life or during the prepuberal period can be complicated.
Which anomalies the Authors are referring to? References are missing TDS Bonde, J. P., Flachs, E.M., Rimborg, S., Glazer, C. H., Giwercman, A., Ramlau‐ Hansen, C. H., … Bräuner, E. V. (2016). The epidemiologic evidence linking prenatal and postnatal exposure to endocrine disrupting chemicals with male reproductive disorders: A systematic review and meta‐analysis. Human Reproduction Update, 23(1), 104–125.

Lines 245-248 The effects can include inducing low sperm count, testicular cancer, cryptorchidism, undescended testis, ambiguous genitalia, "testis dysgenesis syndrome (TDS)": environmental exposure is the primary factor involved in the phenotypes associated with this syndrome.
All these features can be due to genetic defects.
Environmental exposure to mutagens can induce germinal DNA mutations regarding either small DNA sequences or gross chromosomal changes.There is a higher rate of mutation in spermatogenesis in comparison to oogenesis. This is due to the high number of mitosis occurring in male gametogenesis.
The sentence should be rephrased.Done See ref BONDE and TDS
Line 252: “faced with” should be substituted with “facing”DONE
Line 253: “is paralleled with” should be substituted with “parallels”. DONE

Line 257: “methylation tags “ should be substituted with “DNA methylation”
Lines 261-262: For example, major methylation anomalies are observed in Turner and Klinefelter syndromes, although this may be an independent
Removed as well as the corresponding refs ex 48-49
This sentence should be modified or removed. The problem in these syndromes is the chromosome X aneuploidy. The aneuploidy causes no pairing or anomalous sex chromosome pairing at meiosis, which interferes with the correct progression of gametogenesis.
Lines 268-269: It is now recognized that pathologies can be transmitted across generations via epimutations that occur in adult gametogenesis.
This is an overstatement.See Mannikam et al.
Lines 276-277: The oocyte has a redundant but finite capacity for DNA repair, which decreases with maternal age.
This is not proven. There are significantly fewer DNA mutation in oogenesis than in spermatogenesis.
See Lopes and Jurisicova , DNA decays equally share
DNA damage and repair in human oocytes and embryos: a review.Ménézo Y, Dale B, Cohen M.Zygote. 2010 Nov;18(4):357-65.

Expression profiling of DNA repair genes in human oocytes and blastocysts using microarrays.Jaroudi S, Kakourou G, Cawood S, Doshi A, Ranieri DM, Serhal P, Harper JC, SenGupta SB.Hum Reprod. 2009 Oct;24(10):2649-55

Gamete-specific DNA fragmentation in unfertilized human oocytes after intracytoplasmic sperm injection.Lopes S, Jurisicova A, Casper RF.Hum Reprod. 1998 Mar;13(3):703-8.
Expression profile of genes coding for DNA repair in human oocytes using pangenomic microarrays, with a special focus on ROS linked decays.Menezo Y Jr, Russo G, Tosti E, El Mouatassim S, Benkhalifa M.J Assist Reprod Genet. 2007 Nov;24(11):513-20.
Expression profiling of DNA repair genes in human oocytes and blastocysts using microarrays.Jaroudi S, Kakourou G, Cawood S, Doshi A, Ranieri DM, Serhal P, Harper JC, SenGupta SB.Hum Reprod. 2009 Oct;24(10):2649-55. added
Lines 290-292: Plastic EDCs exert a transgenerational effect: they induce oxidative stress (figure 4) via the estrogen receptor (ER), proliferation of activated peroxisome receptors and constitutive androstane receptor (CAR), pregnan X receptor (PXR) and aryl carbon (Ah) receptors.
This statement should be supported by experiments or epidemiology, or both.See Mannikam
The non-genomic effects of endocrine-disrupting chemicals on mammalian sperm.Tavares RS, Escada-Rebelo S, Correia M, Mota PC, Ramalho-Santos J.Reproduction. 2016 Jan;151(1):R1-R13.

Lines 307-308: The spermatozoon allows ancestral history to be transmitted via methylation, as demonstrated by the clear example of the 1944-45 Dutch Famine. There is more and more evidence for this .Ox stress and transmission and the reverse via IGF
Oxidative stress in mouse sperm impairs embryo development, fetal growth and alters adiposity and glucose regulation in female offspring. Lane M, McPherson NO, Fullston T, Spillane M, Sandeman L, Kang WX, Zander-Fox DL.PLoS One. 2014 Jul 9;9(7):e100832. added
It is like transmission of cancer via fatherThis is contradictory to the following remarkof the reviewer…. The global result is a transmission in the next generation. And this is done so far, via the gametes.
On this topic there are other opinions. There is no demonstration of changes in DNA methylation in spermatozoa. The sentence should be modified.
The Effect of Endocrine Disruptors and Environmental and Lifestyle Factors on the Sperm Epigenome Viviane Santana, Albert Salas-Huetos, Emma R. James,and Douglas T. Carrellp
www.cambridge.org/core/terms. https://doi.org/10.1017/9781108762571.004.

Endocrine disruptor induction of epigenetic transgenerational inheritance of disease.Skinner MK.Mol Cell Endocrinol. 2014 Dec;398(1-2):4-12.
Environment, epigenetics and reproduction. Skinner MK.Mol Cell Endocrinol. 2014 Dec;398(1-2):1-3
2 refs added

Lines 308-309: Individuals exposed prenatally to food shortage were found to have reduced methylation status of the imprinted IGF2 gene several decades later.
This has been related to the intrauterine environment. The sentence should be modified.
Lines 314-316: Methylation anomalies, either hypomethylation or hypermethylation, depending upon the target (genes or their promoters), may lead to testicular cancer, initiated from the period of gonadal 315 development.
This is an overstatement. Methylation changes in cancer are known and are related to cancer development in general. The sentence should be modified. Again ART is a good exemple of thisaspect; methylation problems and cancer.Clear evisence for retinoblatome and this last paper.
Assessment of Birth Defects and Cancer Risk in Children Conceived via In Vitro Fertilization in the US.Luke B, Brown MB, Nichols HB, Schymura MJ, Browne ML, Fisher SC, Forestieri NE, Rao C, Yazdy MM, Gershman ST, Ethen MK, Canfield MA, Williams M, Wantman E, Oehninger S, Doody KJ, Eisenberg ML, Baker VL, Lupo PJ.JAMA Netw Open. 2020 Oct 1;3(10):e2022927.
Ref added
Oxidative Stress and Polymorphism in MTHFR SNPs (677 and 1298) in Paternal Sperm DNA is Associated with an Increased Risk of Retinoblastoma in Their Children: A Case-Control Study.Bisht S, Chawla B, Dada R.J Pediatr Genet. 2018 Sep;7(3):103-113
Lines 111-112: Folates are central to a system that involves high molecular trafficking [3]
Other references are necessary/ there are not. It is based on our microarrays studies and other works of our team.But Enciso rfe is OK as well (cited)

Enciso M, Sarasa J, Xanthopoulou L, Bristow S, Bowles MF, Fragouli E, Delhanty J, Wells D. Polymorphisms in the MTHFR gene influence embryo viability and the incidence of aneuploidy. Hum. Genet. 2016, 135: 555–568.

References should be introduced for sentences at lines 88-92, 103-104, 106-108, 108, 112-113, 117-119, 119-120, 126-127, 127-128, 128-130, 161-162, 230-232. DONE

Reviewer 3 Report

This is a descriptive review focused on the impact of assisted reproductive technology and endocrine disruptors (EDCs) on epigenesis and the imprinting process during different stages of development in life.

  1. Overall it's a well-written review with some minor spelling mistakes or typos error. For example: Line 19-the word "physiologie" need to be replaced by physiology, Line 84- "mRNAS" need to "rewritten" as mRNAs.MTHFR needs to be spelled out when writing the first time.
  2.  All the references are recent and cited appropriately.
  3. The only concern I have with this review is that the Authors have described how ARTs or endocrine disruptors can alter imprinting or epigenesis in different reproductive cells and developmental stage in life but has barely focused on how the information provided in the review can help us improving the ART system, what should be the direction of future research to address these issues, what kinds of changes need to be implemented during ART protocol or procedure to minimize the risk of abnormal epigenesis or imprinting. 

Author Response

Reviewer 3 Review Report This is a descriptive review focused on the impact of assisted reproductive technology and endocrine disruptors (EDCs) on epigenesis and the imprinting process during different stages of development in life. 1. Overall it's a well-written review with some minor spelling mistakes or typos error. For example: Line 19-the word "physiologie" need to be replaced by physiology, Line 84- "mRNAS" need to "rewritten" as mRNAs. MTHFR needs to be spelled out when writing the first time. Done
2. All the references are recent and cited appropriately.
3. The only concern I have with this review is that the Authors have described how ARTs or endocrine disruptors can alter imprinting or epigenesis in different reproductive cells and developmental stage in life but has barely focused on how the information provided in the review can help us improving the ART system, what should be the direction of future research to address these issues, what kinds of changes need to be implemented during ART protocol or procedure to minimize the risk of abnormal epigenesis or imprinting. See ref Menezo and Elder 2020 (ref2) and Menezo Clement Dale 2019 The real problem is how the media manufactures fall in thisterrible idea of Essential amino acids removal during the first embryo developmental stages: this means Methionine: and shortage of SAM and cysteine: precursor of glutathione; Terrible. Moreover SAM is a stochiometric activatorsof BHMT for glutathione synthesis
A paper is in project